# Subset Analysis for Screening Drug–Drug Interaction Signal Using Pharmacovigilance Database

**DOI:** 10.3390/pharmaceutics12080762

**Published:** 2020-08-12

**Authors:** Yoshihiro Noguchi, Tomoya Tachi, Hitomi Teramachi

**Affiliations:** 1Laboratory of Clinical Pharmacy, Gifu Pharmaceutical University, 1-25-4, Daigakunishi, Gifu-shi, Gifu 501-1196, Japan; tachi@gifu-pu.ac.jp; 2Laboratory of Community Healthcare Pharmacy, Gifu Pharmaceutical University, Daigakunishi, Gifu-shi, Gifu 501-1196, Japan

**Keywords:** subset analysis, signal detection algorithms, drug-drug interaction, spontaneous reporting systems

## Abstract

Many patients require multi-drug combinations, and adverse event profiles reflect not only the effects of individual drugs but also drug–drug interactions. Although there are several algorithms for detecting drug–drug interaction signals, a simple analysis model is required for early detection of adverse events. Recently, there have been reports of detecting signals of drug–drug interactions using subset analysis, but appropriate detection criterion may not have been used. In this study, we presented and verified an appropriate criterion. The data source used was the Japanese Adverse Drug Event Report (JADER) database; “hypothetical” true data were generated through a combination of signals detected by three detection algorithms. The accuracy of the signal detection of the analytic model under investigation was verified using indicators used in machine learning. The newly proposed subset analysis confirmed that the signal detection was improved, compared with signal detection in the previous subset analysis, on the basis of the indicators of *Accuracy* (0.584 to 0.809), *Precision* (= *Positive predictive value*; *PPV*) (0.302 to 0.596), *Specificity* (0.583 to 0.878), *Youden’s index* (0.170 to 0.465), *F*-*measure* (0.399 to 0.592), and *Negative predictive value* (*NPV*) (0.821 to 0.874). The previous subset analysis detected many false drug–drug interaction signals. Although the newly proposed subset analysis provides slightly lower detection accuracy for drug–drug interaction signals compared to signals compared to the Ω shrinkage measure model, the criteria used in the newly subset analysis significantly reduced the amount of falsely detected signals found in the previous subset analysis.

## 1. Introduction

Drug-induced adverse events (AEs) caused by individual drugs and drug combinations not only hinder treatment but also cause new health hazards. To alleviate this problem, AEs caused by individual drug candidates are closely monitored and investigated during the drug development and approval process [1]. Pre-marketing randomized clinical trials are performed under certain conditions associated with age, gender, and co-morbidities, and some AEs may not be detected. In particular, in pre-marketing randomized clinical trials, patients on combination therapy are usually excluded because the focus is to establish the safety and efficacy of single drugs and not to investigate drug–drug interactions [2]. However, in the real world, many patients suffer from a variety of co-morbidities and use a number of drugs to treat them. The concomitant use of two or more drugs increases the risk of AEs due to drug–drug interactions; the proportion of such AEs is estimated to be up to 30% of unexpected AEs [3].

Therefore, in order to use drugs appropriately in the real world, it is important to understand, in advance, the AEs caused by drug–drug interactions. Post-marketing analysis of AE reports could significantly contribute to the discovery of AEs caused by single drugs or drug–drug interactions that could not be detected before marketing.

For the safety surveillance of drugs, AE reports collected post-marketing are maintained by regulatory agencies as a spontaneous reporting system. There are several algorithms for detecting adverse event signals using the spontaneous reporting system [4]. Of these, the algorithms commonly used for quantitative signal detection include the proportional reporting ratio (PRR) [5], the reporting odds ratio (ROR) [6], the Bayesian confidence propagation neural network (BCPNN) [7], and the empirical Bayesian geometric mean (EBGM) [8].

Additionally, multiple statistical algorithms have been proposed for detecting drug–drug interaction signals [9,10]. However, calculation of the PRR, similar to the risk ratio, and the ROR, similar to the odds ratio, is simple, but that of other algorithms (particularly the algorithm for detecting drug–drug interaction signals) is very complicated.

Therefore, in order to detect the drug–drug interaction signals between *drug D*_1_ and *drug D*_2_, the subset analysis that detects the signal of *drug D*_1_ using the ROR, which is easy to calculate in a subset of patient groups, is often reported [11,12,13].

Of previous studies, several [11,12] have used animal experiments and/or pharmacological data to ensure signal reliability, but the signals obtained with this analysis model are not strictly drug–drug interaction signals; they only showed the effect of drug combinations for the following two reasons [14]:

1.The subset analysis used in this study detects signals from the target AE when the patient group using *drug D*_1_ takes *drug D*_2_. In all patient groups, when the signal value of the target AE is large for drug D_2_, the signal is detected regardless of whether the patient group is using *drug D*_1_.2.Target AE signal intensities when a patient group using *drug D*_1_ takes *drug D*_2_ vs. that when a patient group using *drug D*_2_ takes *drug D*_1_ do not necessarily match. In other words, the value to be adopted as the target AE signal value when *drug D*_1_ and *drug D*_2_ are used concomitantly has not been fixed (i.e., no clear detection criteria have been defined for detecting drug–drug interaction signals).

On the other hand, because the ROR, often used in subset analysis, is easy to calculate, if these shortcomings are improved and the appropriate detection criterion can be set, it might lead to early detection of AEs caused by drug–drug interactions.

In this study, we proposed a new detection criterion for the subset analysis (the newly proposed subset analysis) and verified the detection power using the spontaneous reporting system.

## 2. Materials and Methods

The design of this study is based on a previous paper that discussed trends in methods to detect the signals of AEs caused by individual drugs [15] or drug–drug interactions [14].

### 2.1. Data Sources

The validation dataset was created from the Japanese Adverse Drug Event Report database (JADER), using data from the first quarter of 2004 to the fourth quarter of 2015. The JADER consists of four comma-separated values (csv) files as data tables: DEMO.csv (patient information), DRUG.csv (medicine information), HIST.csv (patient past history), and REAC.csv (AE event information). This study used 374,327 cases registered in the verification dataset.

However, the Japanese authority, the Pharmaceuticals and Medical Devices Agency (PMDA), which owns these data, does not permit sharing the data directly. Therefore, we do not own the JADER. It can be accessed directly here: [http://www.info.pmda.go.jp/fukusayoudb/CsvDownload.jsp] (in Japanese only).

### 2.2. Definitions of Adverse Drug Events

The drugs targeted for the survey are all registered and classified as “suspect drugs” in the validation dataset. The AEs in JADER are based on the preferred terms (PTs) in the Medical Dictionary for Regulatory Activities Japanese version (MedDRA/J). The AE targeted for this study was Stevens–Johnson syndrome (SJS), which was extracted from the dataset using the PT in MedDRA/J. However, the choice of target adverse events is the same as in previous similar studies [14,15], and there was no medical or pharmacological reason for this choice.

### 2.3. “Hypothetical” True Data of Adverse Events for Comparative Verification

The signals obtained from the spontaneous reporting system including the JADER used in this study included unknown AEs that were also detected, which needs to be verified in order to confirm they were true AEs. Moreover, the information provided by the regulatory authorities, of course, does not include unknown AEs. That is, there are no “real” true data for every AE. Therefore, we set “hypothetical” true data because we cannot use “real” true data for validation in this study.

To verify the power of the subset analysis, we prepared “hypothetical” true data of AEs. To generate “hypothetical” true data, we excluded the Ω shrinkage measure model [16] that detected the most conservative signal and the combination risk ratio model [17], which would not detect a signal with a small number of reports, from the five detection algorithms. That is, this study used the combination of signals detected by three algorithms (the additive model [18], the multiplicative model [18], and the chi-square statistical model [19]) as “hypothetical” true data.

### 2.4. Statistical Models and Criteria

#### 2.4.1. Subset Analysis

To detect the signal for the interaction between *drug D*_1_ and *drug D*_2_, we created subsets of the patient group using *drug D*_1_ (or the patient group using *drug D*_2_) (Table 1).

The following equations (Equations (1) and (2)) were used to calculate the ROR and 95% confidence interval (95% CI) of the target AE caused by *drug D*_1_ (or *drug D*_2_) from the generated subset, respectively. For the signal of a patient group on *drug D*_1_ that takes *drug D*_2_, the number of each report can be expressed as follows: *N*_11_ = *n*_111_, *N*_10_ = *n*_110_, *N*_01_ = *n*_101_, *N*_00_ = *n*_100_. On the other hand, for the signal of a patient group on *drug D*_2_ that takes *drug D*_1_, the number of each report can be expressed as follows: *N*_11_ = *n*_111_, *N*_10_ = *n*_110_, *N*_01_ = *n*_011_, *N*_00_ = *n*_010_.
(1)$$\mathrm{ROR}\;=\frac{N_{11}/N_{00}}{N_{01}/N_{10}}\;$$
(2)$$\mathrm{ROR}\;\left(95\%\;\mathrm{CI}\right)=e^{\ln\left(\mathrm{ROR}\right)\;\pm 1.96\sqrt{\frac{1}{N_{11}}+\frac{1}{N_{10}}+\frac{1}{N_{01}}+\frac{1}{N_{00}}}}$$

In previous studies [11,12,13], if the signal for *drug D*_2_ was detected in the subset of a patient group using *drug D*_1_ or if the signal for *drug D*_1_ was detected in the subset of a patient group using *drug D*_2_, this signal was considered the drug–drug interaction signal. The criterion that a signal only needs to be detected from a subset of either patient group is ambiguous, highlighting the two shortcomings mentioned earlier. Therefore, for the newly proposed subset analysis, a case was redefined as the drug–drug interaction signal if a signal was detected in both subsets of a patient group using *drug D*_1_ and a patient group using *drug D*_2_.

#### 2.4.2. Ω Shrinkage Measure Model

The Ω shrinkage measure model [16] is based on a measure calculated as the ratio of the observed reporting ratio of the AE associated with the combination of two drugs and its expected value; this model is used by the Uppsala Monitoring Center (UMC) and the World Health Organization (WHO) Collaborating Centre for International Drug Monitoring for signal analysis of drug–drug interactions (Table 1, Equations (3)–(7)).
(3)$$\Omega \;=log_{2}\frac{n_{111}+0.5}{E_{111}+0.5}\;$$
where *n*_111_ is the reported number of AEs caused by the combination of two drugs, and *E*_111_ is the expected value of AEs caused by the combination of two drugs.

*ϕ*(0.975) is 97.5% of the standard normal distribution and Ω_025_ > 0 is used as a threshold to screen for signals under the combination of two drugs (Equation (4)).
(4)$$\Omega_{025}=\;\Omega \;-\frac{\phi \left(0.975\right)}{log\left(2\right)\sqrt{n_{111}}}$$

To calculate *E*_111_, we used the following Equations (5)–(7).
(5)$$f_{00}=\frac{n_{001}}{n_{00+}},f_{10}=\frac{n_{101}}{n_{10+}},f_{01}=\frac{n_{011}}{n_{01+}},f_{11}=\frac{n_{111}}{n_{11+}}$$
(6)$$g_{11}=1-\frac{1}{\max\left(\frac{f_{00}}{1-f_{00}},\;\frac{f_{10}}{1-f_{10}}\right)+\max\left(\frac{f_{00}}{1-f_{00}},\;\frac{f_{01}}{1-f_{01}}\right)-\frac{f_{00}}{1-f_{00}}+1\;}\;$$

When *f*_10_ < *f*_00_ (which denotes no risk of AE caused by drug *D*_1_), the most sensible estimator *g*_11_ = max (*f*_00_, *f*_01_) is yielded and vice versa when *f*_01_ < *f*_00_.

Norén et al. re-expressed the observed and expected RRR *f*_11_ and g_11_ in terms of the observed number of reports n_111_ and expected numbers of reports *E*_111_ = *g*_11_ × *n*_11+_, respectively:(7)$$\frac{f_{11}}{g_{11}}=\frac{n_{111}/n_{11+}}{E_{111}/n_{11+}}=\frac{n_{111}}{E_{111}}$$

### 2.5. Evaluation of Detection Models

#### 2.5.1. Using Evaluations of Classification in Machine Learning

The evaluation indicators that we set were *Accuracy* (Table 2, Equation (8)), *Precision* (*Positive predictive value*; *PPV*) (Table 2, Equation (9)), *Recall* (*Sensitivity*) (Table 2, Equation (10)), *Specificity* (Table 2, Equation (11)), *Youden’s index* (Table 2, Equation (12)), *F*-measure (Table 2, Equation (13)), and *Negative predictive value* (*NPV*) (Table 2, Equation (14)).
(8)$$Accuracy=\frac{TP+TN}{TP+FP+TN+FN}$$
(9)$$Precision\;\left(Positive\;predictive\;value;PPV\right)=\frac{TP}{TP+FP}$$
(10)$$Recall\;\left(Sensitivity\right)=\frac{TP}{TP+FN}$$
(11)$$Specificity=\frac{TN}{FP+TN}$$
(12)$$Youden^{\prime }s\;index=Sensitivity+Specificity-1$$
(13)$$F-measure=\frac{2\times Recall\times Precision}{Recall+Precision}$$
(14)$$Negative\;predictive\;value\;\left(NPV\right)=\frac{TN}{TN+FN}$$

#### 2.5.2. Cohen’s Kappa Coefficient

The commonality of the signals detected by each statistical model was evaluated using *Cohen’s kappa coefficient* (*κ*), proportionate agreement for positive rating (*P*_positive_), and proportionate agreement for negative rating (*P*_negative_), as reported in a previous study [14,15]. In this study, we investigated the similarities with Ω shrinkage measure model for the previous/newly proposed subset analysis.

### 2.6. Analysis Software

The analysis software in this study used Visual Mining Studio (NTT DATA Mathematical Systems Inc., Shinjuku-ku, Tokyo, Japan) version 8.4 and Microsoft Excel 2019 (Microsoft Corp., Redmond, WA, USA).

## 3. Results

### 3.1. Evaluations of Classification in Machine Learning

Among all 374,327 cases analyzed, there were 3924 *drug D*_1_–*drug D*_2_–SJS combinations. Of these, 923 combinations were detected by all three algorithms—the additive model [18], the multiplicative model [18], and the chi-square statistics model [19]. In this study, these combinations were treated as “hypothetical” true data.

The evaluation of the analysis model is shown in Table 3 and Table 4.

Table 3 shows the number of True positive (TP), False Positive (FP), True Negative (TN), and False Negative (FN).

A total of 1793 combinations were detected by the previous subset analysis (*True positive*: 542, *False positive*: 1251). On the other hand, the newly proposed subset analysis detected 909 combinations of signals (*True positive*: 542, *False positive*: 367) (Table 3).

The detection accuracy shown in Table 4 was calculated from the values shown in Table 3.

In addition, the newly proposed subset analysis confirmed that the signal detection was improved with respect to the indicators of *Accuracy* (0.584 to 0.809), *Precision* (*PPV*) (0.302 to 0.596), *Specificity* (0.583 to 0.878), *Youden’s index* (0.170 to 0.465), *F*-*measure* (0.399 to 0.592), and *NPV* (0.821 to 0.874) as compared with the signal detection in the previous subset analysis (Table 3).

The values of each indicator of the Ω shrinkage measure model were *Accuracy* (0.858), *Precision* (PPV) (0.756), *Recall* (*Sensitivity*) (0.583), *Specificity* (0.942), *Youden’s index* (0.525), *F*-*measure* (0.658), and *NPV* (0.880) (Table 4).

### 3.2. Cohen’s Kappa Coefficient

The similarity between the detection results of the Ω shrinkage measure model and that of the newly proposed subset analysis was *κ* (95% CI): 0.375 (0.355–0.395), *P*_positive_: 0.502, and *P*_negative_: 0.870. The similarity was *κ* (95% CI): 0.355 (0.327–0.384), *P*_positive_: 0.678, and *P*_negative_: 0.674 when targeting three or more reports (Table 5).

## 4. Discussions

In this study, we evaluated the accuracy of drug–drug interaction signals for the newly proposed subset analysis that modified two shortcomings of the previous subset analysis on the basis of data from the spontaneous reporting system.

There were 3924 pairs of *drug D*_1_–*drug D*_2_–SJS in the spontaneous reporting system, JADER. There are several known combinations of drugs that onset SJS by drug–drug interactions [20]. On the other hand, there are some combinations that have not yet been reported. Recently, we used the Ω shrinkage measure model to report potential drug combinations for the onset of SJS in concomitant use with antiepileptic drugs [21]. Not all AEs have been identified and there are still many unknown AEs. Unfortunately, unknown AE data do not exist anywhere in the world; there were no “real” true data for AEs. Therefore, to verify the accuracy of the subset analysis, we needed to prepare “hypothetical” true data of AEs. A previous comparative study [14] of five algorithms for detecting drug–drug interaction signals revealed that the Ω shrinkage measure model [16] detected the most conservative signal, while the combination risk ratio model [17] did not detect any interaction signal in less than three reports due to the detection criterion. Therefore, of the five algorithms, we used the combination of signals detected by the three algorithms (the additive model, the multiplicative model, and the chi-square statistical model) as “hypothetical” true data.

Among the previous subset analysis, the newly proposed subset analysis, and the Ω shrinkage measure model, most signals were detected by the previous subset analysis with 1793 pairs (45.7% of the total combinations, *Accuracy*: 0.584, *Precision* (*PPV*): 0.302, *Recall* (*Sensitivity*): 0.587, *Specificity*: 0.583, *Youden’s index*: 0.170, *F*-*measure*: 0.399, and *NPV*: 0.821), followed by the newly proposed subset analysis with 909 pairs (23.2% of the total combinations, *Accuracy*: 0.809, *Precision* (*PPV*): 0.596, *Recall* (*Sensitivity*): 0.587, *Specificity*: 0.878, *Youden’s index*: 0.465, *F*-*measure*: 0.592, and *NPV*: 0.874). In contrast, the Ω shrinkage measure model detected the fewest signals with 712 pairs (18.1% of the total combinations, *Accuracy*: 0.858, *Precision* (*PPV*): 0.756, *Recall* (*Sensitivity*): 0.583, *Specificity*: 0.942, *Youden’s index*: 0.525, *F*-*measure*: 0.658, and *NPV*: 0.880) (Table 2, Table 4).

This result indicates that the accuracy of signal detection has been greatly improved in the newly proposed subset analysis with a simple modification of the previous subset analysis. However, the newly proposed subset analysis exhibited slightly lower power and accuracy for detecting the drug–drug interaction signals compared to the Ω shrinkage measure model.

Verification by the number of reports showed that when the number of reports (*N*_11_; *n*_111_) < 2, the accuracy (*Youden’s index*, *F*-*measure*) of signal detection was higher in the newly proposed subset analysis than in the Ω shrinkage measure model (*Youden’s index*: the newly proposed subset analysis (0.337) vs. the Ω shrinkage measure model (0.174), *F*-*measure*: the newly proposed subset analysis (0.448) vs. the Ω shrinkage measure model (0.298)).

However, as the number of reports increased, the Ω shrinkage measure model became more accurate (*Youden’s index*: the newly proposed subset analysis (0.465) vs. the Ω shrinkage measure model (0.525), *F*-*measure*: the newly proposed subset analysis (0.592) vs. the Ω shrinkage measure model (0.658)) (Table 4).

Additionally, the *True positive* values for the previous subset analysis and the newly proposed subset analysis were the same (Table 3). Since all signals obtained by the newly proposed subset analysis were included in the previous subset analysis, this result indicates that the detection criterion of the previous subset analysis was loose and that the data contained false positives.

The similarity between the newly proposed subset analysis and the Ω shrinkage measure model was *κ* (95% CI): 0.375 (0.355–0.395), *P*_positive_: 0.502, and *P*_negative_: 0.870. On the other hand, the similarity between the previously subset analysis and the Ω shrinkage measure model was *κ* (95% CI): 0.088 (0.071–0.105), *P*_positive_: 0.325, and *P*_negative_: 0.684. Thus, the newly proposed subset analysis w more similar to the Ω shrinkage measure model than the previously subset analysis. However, the similarity of the newly proposed subset analysis and the Ω shrinkage measure model is not very high. Additionally, when the number of reports (*N*_11_; *n*_111_) was ≥3, no significant change was observed in the similarity between the Ω shrinkage measure model and the newly proposed subset analysis. Despite not being similar to the Ω shrinkage model, the newly subset analysis showed a high degree of accuracy. This result suggests that the newly subset analysis may be detecting signals that the Ω shrinkage model has failed to detect.

This study has the following three limitations. First, unfortunately, unknown AE data do not exist anywhere in the world [14]. Therefore, there were no “real” true data for AEs. Thus, for the purpose of verification, it was necessary to set “hypothetical” true data for AEs instead of “real” true data. Therefore, of the five algorithms for detecting drug–drug interaction signals, we used the combination of signals detected by the three algorithms (the additive model, the multiplicative model, and the chi-square statistical model) as “hypothetical” true data in this study. In other words, the hypothetical true data consisted of statistically based *drug D*_1_–*drug D*_2_–AE combinations, not pharmacologically based combinations.

Second, usually it is important to compare detection trends using all AEs recorded in the validation dataset created on the basis of a spontaneous reporting system; however, it takes an extremely long time to calculate signal values for all combinations of drug–drug interactions. Such a study design is not realistic. Therefore, this study targeted SJS, the same AE used in previous comparative studies [14,15]; if different reference sets were used, the possibility of obtaining different performance characteristics might not be ruled out. There are fewer enrolled cases than in the global dataset because JADER is limited to cases in Japan. However, the signal detection is based on a comparison between the ratio of reported cases (*N*) to expected values (*E*). Therefore, differences in the number of cases enrolled in the spontaneous reporting system had only a very small statistical impact in this study. Recently, validation of the number of cases enrolled in the spontaneous reporting system has also been reported by Caster et al. [22]. Moreover, differences in the way regulatory authorities think may result in a different tendency to register AEs to the spontaneous reporting system. For example, the Food and Drug Administration Adverse Events Reporting System (FAERS) in the United States has also registered reports from non-medical professionals, but JADER has not registered reports from patients until recently. It is unknown how the differences in registration tendencies affect the results of this study.

Finally, neither the general algorithms for detecting drug–drug interaction signals nor the proposed subset analysis in this study were antagonistic; only signals of synergistic interactions were detected [10].

## 5. Conclusions

In recent years, the need for safety signal screening has been demanded, not only for single drugs but also for drug–drug interactions. Although several methods for detecting signals of drug interaction have been reported, it is difficult to say that these methods are used because many of them are complicated in calculation. Therefore, there were several cases [11,12,13] where subset analysis using the algorithm for detecting signals of single drugs (e.g., ROR [6]) was used for signal detection of drug–drug interactions before its validity was verified.

This study showed that there were many false positives in the existing subset analysis, albeit under limited conditions. Additionally, very simple modifications of the detection criteria were made to solve two problems associated with the previous subset analysis for exploring recently reported drug–drug interaction signals. This modification helped to reduce falsely detected signals found in the previous subset analysis.

Moreover, the newly proposed subset analysis is more similar to the Ω shrinkage measure model than the previous subset analysis, but the similarity with the Ω shrinkage measure model is not as high. However, the newly proposed subset analysis showed that although the detection accuracy of the drug–drug interaction signal was slightly lower than that of the Ω shrinkage measure model, the detection accuracy was sufficient. This result may also indicate the possibility of detecting signals that cannot be detected by the Ω shrinkage measure model.

## Figures and Tables

**Table 1 pharmaceutics-12-00762-t001:** The 4 × 2 contingency table for signal detection: AE: adverse event; n: the number of reports (e.g., *n*_+++_: the number of all reports, *n*_111_: the number of *drug D*_1_- and *drug D*_2_-induced target AE reports).

	Target AE	Other AEs	Total
*Concomitant use* of *drug D_1_* and *drug D_2_*	*n* _111_	*n* _110_	*n* _11+_
only *drug D_1_*	*n* _101_	*n* _100_	*n* _10+_
only *drug D_2_*	*n* _011_	*n* _010_	*n* _01+_
Neither *drug D_1_* or *drug D_2_*	*n* _001_	*n* _000_	*n* _00+_
Total	*n* _++1_	*n* _++0_	*n* _+++_

**Table 2 pharmaceutics-12-00762-t002:** Agreement between the criterion A and the “hypothetical” true data. AE: adverse event, *TP*: *True Positive*, *FP*: *False Positive*, *FN*: *False Negative*, *TN*: *True Negative*.

		*“Hypothetical” True Data*	
		AE	non-AEs
***analysis model***	signal	*TP*	*FP*
	Non-signal	*FN*	*TN*

**Table 3 pharmaceutics-12-00762-t003:** The number of True positive, False positive, True negative, and False negative.

Analysis Model	*TP*	*FP*	*TN*	*FN*
Previous subset analysis	542	1251	1750	381
Newly proposed subset analysis	542	367	2634	381
Ω shrinkage measure model	538	174	2827	385

TP: True positive, FP: False positive, TN: True negative, FN: False negative.

**Table 4 pharmaceutics-12-00762-t004:** Evaluation of detected drug–drug interaction signals.

**Analysis Model**	***Accuracy***	***Precision*** **(*PPV*)**	***Recall*** **(*Sensitivity*)**	***Specificity***	***Youden’s Index***	***F*** **-*Measure***	***NPV***
Previous subset analysis	0.584	0.302	0.587	0.583	0.170	0.399	0.821
Newly proposed subset analysis	0.809	0.596	0.587	0.878	0.465	0.592	0.874
Ω shrinkage measure model	0.858	0.756	0.583	0.942	0.525	0.658	0.880

*PPV: Positive predictive value, NPV: Negative predictive value.*

**Table 5 pharmaceutics-12-00762-t005:** The similarity between the Ω shrinkage measure model and subset analysis.

Analysis Model	All Case			*n*_111_ ≥ 3		
	*κ* (*95% CI*)	*P* _positive_	*P* _negative_	*κ* (*95% CI*)	*P* _positive_	*P* _negative_
Previous subset analysis	0.088 (0.071–0.105)	0.325	0.684	−0.120 (−0.151–0.088)	0.556	0.296
Newly proposed subset analysis	0.375 (0.355–0.395)	0.502	0.870	0.355 (0.327–0.384)	0.678	0.674

*n*_111_: targeting three or more reports, *κ*: *Cohen’s kappa coefficient*, *P*_positive_: proportionate agreement for positive rating, *P*_negative_: proportionate agreement for negative rating.

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
