# Peer review of "Subset Analysis for Screening Drug–Drug Interaction Signal Using Pharmacovigilance Database"

_pharmaceutics, 2020, doi:10.3390/pharmaceutics12080762_

Round 1

Reviewer 1 Report

The manuscript has been written very organised way. The authors explained new detection criterion for the subset analysis in a scientific way. They sum up the conclusion, but it can be improved.

The manuscript has been written based on reported case (not real data). The authors address the limitation in their discussion section. However, if the authors can give some statements at conclusion section considering the aspect linking with previous published reference that would add extra value to the readers.  Overall, the authors explained the manuscript very well organised way.

Author Response

Thank you for reviewing our paper.

We added the reason why we did not use real true data in this study.

  Line 244-248: There are several known combinations of drugs that onset SJS by drug-drug interactions [20]. On the other hand, there are some combinations that have not yet been reported. Recently, we have used the Ω shrinkage measure model to report potential drug combinations for the onset of SJS in concomitant use with antiepileptic drugs [21]. Not all AEs have been identified and there are still many unknown AEs. Unfortunately, unknown~

[20] Cheng, F.J.; Syu, F.K.; Lee, K.H.; Chen, F.C.; Wu, C.H.; Chen, C.C. Correlation between drug-drug interaction-induced Stevens-Johnson syndrome and related deaths in Taiwan. J Food Drug Anal. 2016, 24, 427-432. doi: 10.1016/j.jfda.2015.11.009.

[21] Noguchi, Y.; Takaoka, M.; Hayashi, T; Tachi, T; Teramachi, H.; Antiepileptic combination therapy with Stevens-Johnson syndrome and toxic epidermal necrolysis: Analysis of a Japanese pharmacovigilance database. Epilepsia. 2020. doi: 10.1111/epi.16626.

Reviewer 2 Report

The manuscript entitled "Subset analysis for screening drug-drug interaction signal using pharmacovigilance database" by Yoshihiro Noguchi, Tomoya Tachi and Hitomi Teramachi, provides interesting results regarding the adverse drugs events. In general, the work is well organized and presented, despite the minor concerns about the methodology. Below, please find questions and additional remarks, which could help to improve the manuscript:

Please, concern below remarks.

1) The authors chosen Stevens-Johnson Syndrome in order to compare their results with the published elsewhere, beside the obvious scientific soundness of the results, what was the driving force to chose SJS? Please provide short description of SJS and its impact on the treatment of the patients.

2) The data source is the PMDA, although I believe that the readers which are not familiar with Japanese or do not have time to download the data, may be curious what is the structure of the data. Please, provide brief description of the csv databases.

3) Please, provide an ICD-10 code for targeted adverse effect? (Is it ICD-10-CM Code L51.1?)

4) Please provide the information what tools/software were used during study.

5) As a result of data mining we are familiarized with raw error measures. I believe that some of the readers would like to see the scripts/procedures (possibly written in FOSS, R/Python/Java), which were used during the study.

6) The Authors have stated that there is no "real" data, and the cases are reported only, therefore it is hard to draw more general conclusions about the source of interactions, but I recommend to compare drug-drug with previous findings (e.g Fu-Jen Cheng, Fei-Kai Syu, Kuo-Hsin Lee, Fu-Cheng Chen, Chien-Hung Wu, Chien-Chih Chen, Correlation between drug–drug interaction-induced Stevens–Johnson syndrome and related deaths in Taiwan, Journal of Food and Drug Analysis, Volume 24, Issue 2, 2016, Pages 427-432).

Regarding all above, I recommend to publish the manuscript with minor revision.

Author Response

Thank you for reviewing our paper.

We made the appropriate corrections as the reviewer pointed out.

1) The authors chosen Stevens-Johnson Syndrome in order to compare their results with the published elsewhere, beside the obvious scientific soundness of the results, what was the driving force to chose SJS? Please provide short description of SJS and its impact on the treatment of the patients.

→The target adverse events (Stevens-Johnson syndrome) were kept the same as in previous similar studies. There was no medical or pharmacological reason for the selection of target adverse events.

  Line 101-102: However, the choice of target adverse events is the same as in previous similar studies [14, 15], and there is no medical or pharmacological reason for this choice.

2) The data source is the PMDA, although I believe that the readers which are not familiar with Japanese or do not have time to download the data, may be curious what is the structure of the data. Please, provide brief description of the csv databases.

→As you pointed out, we have added information in the “2.1. Data sources” section.

  Line 87-89: The JADER consists of four comma-separated values (csv) files as data tables: DEMO.csv (patient information), DRUG.csv (medicine information), HIST.csv (patient past history), and REAC.csv (AE event information).

3) Please, provide an ICD-10 code for targeted adverse effect? (Is it ICD-10-CM Code L51.1?)

→Adverse events registered in JADER are based on Medical Dictionary for Regulatory Activities Japanese version (MedDRA/J). Hence, ICD-10 was not used in this study to extract adverse event cases.

It's hard to understand, so it's been corrected as follows.

Line 97-101: The AEs in JADER are based on the preferred terms (PTs) in the Medical Dictionary for Regulatory Activities Japanese version (MedDRA /J). The AE targeted for this study was Stevens-Johnson syndrome (SJS), which was extracted from the dataset using the PT in MedDRA / J. preferred term (PT) in the Medical Dictionary for Regulatory Activities Japanese version (MedDRA /J).

4) Please provide the information what tools/software were used during study.

→As you pointed out, we have added information on the software used in the “2.1. Data sources” section.

  Line 203-206: 2.6. Analysis software The analysis software in this study used Visual Mining Studio® (NTT DATA Mathematical Systems Inc. Shinjuku-ku,Tokyo, JAPAN) version 8.4 and Microsoft Excel® 2019 (Microsoft Corp., Redmond, WA USA).

5) As a result of data mining we are familiarized with raw error measures. I believe that some of the readers would like to see the scripts/procedures (possibly written in FOSS, R/Python/Java), which were used during the study.

→No script was created to calculate the signal score. A number of research groups have proposed various methods for calculating the signal score that were validated in this study. However, no scripts (e.g., R packages) for these calculations have been presented. Therefore, our group is planning to develop an R package that includes the methods proposed by other groups to calculate the signal score.

6) The Authors have stated that there is no "real" data, and the cases are reported only, therefore it is hard to draw more general conclusions about the source of interactions, but I recommend to compare drug-drug with previous findings (e.g Fu-Jen Cheng, Fei-Kai Syu, Kuo-Hsin Lee, Fu-Cheng Chen, Chien-Hung Wu, Chien-Chih Chen, Correlation between drug–drug interaction-induced Stevens–Johnson syndrome and related deaths in Taiwan, Journal of Food and Drug Analysis, Volume 24, Issue 2, 2016, Pages 427-432).

→We have added the following.

  Line 244-248: There are several known combinations of drugs that onset SJS by drug-drug interactions [20]. On the other hand, there are some combinations that have not yet been reported. Recently, we have used the Ω shrinkage measure model to report potential drug combinations for the onset of SJS in concomitant use with antiepileptic drugs [21]. Not all AEs have been identified and there are still many unknown AEs. Unfortunately, unknown~

[20] Cheng, F.J.; Syu, F.K.; Lee, K.H.; Chen, F.C.; Wu, C.H.; Chen, C.C. Correlation between drug-drug interaction-induced Stevens-Johnson syndrome and related deaths in Taiwan. J Food Drug Anal. 2016, 24, 427-432. doi: 10.1016/j.jfda.2015.11.009.

[21] Noguchi, Y.; Takaoka, M.; Hayashi, T; Tachi, T; Teramachi, H.; Antiepileptic combination therapy with Stevens-Johnson syndrome and toxic epidermal necrolysis: Analysis of a Japanese pharmacovigilance database. Epilepsia. 2020. doi: 10.1111/epi.16626.

Reviewer 3 Report

Manuscript entitled "Subset analysis for screening drug-drug interaction signal using pharmacovigilance database" describes the importance of new subset analysis in identifying the drug-drug interactions with high accuracy, precision, and specificity. While the authors developed a platform that enables better predictions of DDI, it has to be revised before it can be deemed publishable. 

1) Please discuss few real case studies and the impact that it might cause to patients considering the old subset analysis. This creates interests to the readers as the main focus of the manuscript is about drugs and their interactions. I see nothing covered in this regard. Please revise.

2) The existing Ω shrinkage model has better predictability than old and new subset analysis. What features in Ω shrinkage model makes it a better model to predict DDI? Please discuss in detail. In addition, I would like to see a comparative table that describes the key attributes of all the three models. This helps to get a better feel of the models at a quick glance. 

3) Also, these predictions were made from the JADER data alone. What is the opinion of the authors with respect to global importance of this new subset analysis approach. Please clarify.

4) With the Ω shrinkage model already in place and generating better predictions, what is the advantage in using the new subset analysis approach. Please clarify.

Overall, the new subset analysis has made significant progress in enabling better predictions. However, there are few missing facts in the manuscript and I would suggest authors to revise on these aspects. 

Author Response

Thank you for reviewing our paper.

We made the appropriate corrections as the reviewer pointed out.

1) Please discuss few real case studies and the impact that it might cause to patients considering the old subset analysis. This creates interests to the readers as the main focus of the manuscript is about drugs and their interactions. I see nothing covered in this regard. Please revise.

→The main purpose of this study was to revise the previous subset analysis, which had a high number of false positives. We have demonstrated that with an appropriate study design.

Line 224-227: the newly proposed subset analysis confirmed that the signal detection was improved with respect to the indicators of Accuracy (0.584 → 0.809), Precision (PPV) (0.302 → 0.596), Specificity (0.583 → 0.878), Youden’s index (0.170 → 0.465), F-measure (0.399 → 0.592), and NPV (0.821 → 0.874) as compared with the signal detection in the previous subset analysis (Table 1).

We are confident that the publication of this paper will discourage researchers from using the previous subset analysis, which are often false positives.

The pharmacological significance of the detected signals should be examined separately and is outside the main purpose of this paper. If we are able to detect a significant signal, as you point out, we will report on that signal separately in other papers.

2) The existing Ω shrinkage model has better predictability than old and new subset analysis. What features in Ω shrinkage model makes it a better model to predict DDI? Please discuss in detail. In addition, I would like to see a comparative table that describes the key attributes of all the three models. This helps to get a better feel of the models at a quick glance.

→The signal detection is based on a comparison between the ratio of reported cases (N) to expected values (E). All the methods used in this study are based on this principle of disproporotinality. All the three models have the same principles. Therefore, we cannot answer your question. However, we want to emphasize the following to you.

The Ω shrinkage model is designed to detect signals of drug-drug interactions. While the ROR is essentially a method of investigating adverse events caused by a single drug. However, in specific drug use groups as a subset, drug-drug interactions can be also explored by using the ROR.

The conventional method had two problems described in the manuscript.

Line 68-75:

  1. The subset analysis used in this study detects signals from the target AE when the patient group using drug D1 takes drug D2. In all patient groups, when the signal value of the target AE is large for drug D2, the signal is detected regardless of whether the patient group is using drug D1.
  2. Target AE signal intensities when a patient group using drug D1 takes drug D2 vs that when a patient group using drug D2 takes drug D1 do not necessarily match. In other words, the value to be adopted as the target AE signal value when drug D1 and drug D2 are used concomitantly has not been fixed (i.e., no clear detection criteria have been defined for detecting drug-drug interaction signals).

 Our proposed newly subset analysis overcomes these problems.

  Despite not being similar to the Ω shrinkage model, the newly subset analysis has shown a high degree of accuracy.

  This result suggests that the newly subset analysis may be detecting signals that the Ω shrinkage model has failed to detect.

3) Also, these predictions were made from the JADER data alone. What is the opinion of the authors with respect to global importance of this new subset analysis approach. Please clarify.

→As you pointed out, we have added the following.

  Line 303-313: There are fewer enrolled cases than in the global dataset, because JADER is limited to cases in Japan. However, the signal detection is based on a comparison between the ratio of reported cases (N) to expected values (E). Therefore, differences in the number of cases enrolled in the spontaneous reporting system have only a very small statistical impact in this study. Recently, validation of the number of cases enrolled in the spontaneous reporting system has also been reported by Caster et al [22]. While, differences in the way regulatory authorities think may result in a different tendency to register AEs to the spontaneous reporting system. For example, Food and Drug Administration Adverse Events Reporting System (FAERS) in United States has also registered reports from non-medical professionals, but JADER has not registered reports from patients until recently. It is unknown how the differences in registration tendencies affect the results of this study.

4) With the Ω shrinkage model already in place and generating better predictions, what is the advantage in using the new subset analysis approach. Please clarify.

→It is noted in the “5. Conclusion” Section.

Line 336-337: This result may also indicate the possibility of detecting signals that cannot be detected by the Ω-shrinkage measure model.

→We have added to the discussion section as well.

Line 286-292: However, the similarity of the newly proposed subset analysis and the Ω-shrinkage measure model is not very high. Additionally, when the number of reports (N11; n111) was ≥ 3, no significant change was observed in the similarity between the Ω shrinkage measure model and the newly proposed subset analysis. Despite not being similar to the Ω shrinkage model, the newly subset analysis has shown a high degree of accuracy. This result suggests that the newly subset analysis may be detecting signals that the Ω shrinkage model has failed to detect.

Round 2

Reviewer 3 Report

Accepted.